# Risk Assessment of Trigonelline in Coffee and Coffee By-Products

**DOI:** 10.3390/molecules28083460

**Published:** 2023-04-14

**Authors:** Nick Konstantinidis, Heike Franke, Steffen Schwarz, Dirk W. Lachenmeier

**Affiliations:** 1Postgraduate Study Program “Toxicology and Environmental Protection”, Rudolf-Boehm-Institut für Pharmakologie und Toxikologie, Universität Leipzig, Härtelstraße 16–18, 04107 Leipzig, Germany; nick.konstantinidis@web.de (N.K.); heike.franke@medizin.uni-leipzig.de (H.F.); 2Chemisches und Veterinäruntersuchungsamt (CVUA) Karlsruhe, Weissenburger Strasse 3, 76187 Karlsruhe, Germany; 3Coffee Consulate, Hans-Thoma-Strasse 20, 68163 Mannheim, Germany; schwarz@coffee-consulate.com

**Keywords:** trigonelline, coffee, coffee by-products, toxicity, human, risk assessment

## Abstract

Trigonelline is a bioactive pyridine alkaloid that occurs naturally in high concentrations in coffee (up to 7.2 g/kg) and coffee by-products (up to 62.6 g/kg) such as coffee leaves, flowers, cherry husks or pulp, parchment, silver skin, and spent grounds. In the past, coffee by-products were mostly considered waste and discarded. In recent years, however, the use of coffee by-products as food has attracted interest because of their economic and nutritional value and the environmental benefits of sustainable resource use. Their authorization as so-called novel foods in the European Union may lead to increased oral exposure of the general population to trigonelline. Therefore, the aim of this review was to assess the risk to human health of acute and chronic exposure to trigonelline from coffee and coffee by-products. An electronic literature search was performed. Current toxicological knowledge is limited, with few human data available and a lack of epidemiological and clinical studies. There was no evidence of adverse effects after acute exposure. No conclusion can be drawn on chronic exposure to isolated trigonelline due to the lack of data. However, trigonelline ingested as a component of coffee and coffee by-products appears to be safe for human health, based on the safe traditional use of these products.

## 1. Introduction

Trigonelline (IUPAC: 1-methylpyridin-1-ium-3-carboxylate, C_7_H_7_NO_2_) is a solid, crystallizable pyridine alkaloid that was first isolated from the seeds of fenugreek (*Trigonella foenum-graecum* L.) by German pharmacist Ernst Friedrich Jahns in 1885 [1,2,3]. It is a small, highly hydrophilic (Table 1), and thermolabile phytochemical [1,2,3,4,5,6,7,8] naturally occurring in numerous plants as a secondary metabolite [9,10,11,12,13]. Particularly high contents up to 34.2 g/kg are found in green coffee beans (family Rubiaceae, genus *Coffea*) [4,13,14,15,16,17,18,19,20], with trigonelline thus being the most abundant alkaloid in green coffee beans second to caffeine [17]. Apart from plants, trigonelline has been detected in several animals [10,11,21], and is also present in human plasma, serum, and urine [17,22,23,24,25,26].

As discovered in green peas (*Pisum sativum* L.) by Joshi and Handler in 1960, trigonelline is believed to be synthesized *in planta* via methylation of the nitrogen atom of nicotinic acid, with *S*-adenosyl-*L*-methionine (SAM) serving as methyl donor and SAM:nicotinic acid *N*-methyltransferase, synonymously *N*-methyltransferase or trigonelline synthase (EC 2.1.1.7), as catalyzing enzyme (Figure 1) [27,28,29]. In 2014, the genes of this enzyme were successfully isolated from beans of *Coffea arabica* L. [30], suggesting a similar biosynthetic pathway in coffee plants. In these, trigonelline is formed in all tissues (yet, primarily in the fruits (Figure 2)), with higher outputs in younger than in older tissues [9,31,32]. A detailed overview of the biosynthesis in plants is provided by Ashihara et al. [29]. In animals and humans, biosynthesis is still unknown, but may happen analogously via methylation of nicotinic acid [26].

Various functions of trigonelline in planta have been proposed, such as serving as a nutrient source and compatible solute, involvement in cell cycle regulation by arresting the G2 phase, signal transduction, nyctinasty (orientation of a plant’s leaves, flowers, cotyledons, or branches toward the light source to maximize photosynthesis [33]), and nodulation (symbiotic relationship of capable plants with certain bacteria (rhizobia), leading to the formation of specific root structures (nodules) to enable nitrogen fixation [34]) [29,35,36]. The most plausible role may be detoxification of nicotinic acid, which, in excess concentrations, is detrimental to plants [9,13,36,37].

No physiological functions in animals and humans are known [12,38]. However, trigonelline provides nutritional value as it decomposes back to its precursor nicotinic acid (niacin or vitamin B_3_) in coffee beans upon exposure to roasting temperatures (see Section 5) [17,39,40,41]. Furthermore, multiple possible health-promoting effects in humans are documented. These include antimicrobial and anticariogenic [42], antioxidative [43,44,45], antilipidemic [43,46,47], hypocholesterolemic [43], and hypoglycemic effects [46,48,49,50]. Recently, trigonelline was also shown to protect against the formation of kidney stones [51]. The possible health-promoting effects exhibited by trigonelline have been reviewed elsewhere [12,52].

The most important source of trigonelline commercially available is coffee [29]. Originating from the Kaffa highlands of Ethiopia, where coffee is believed to have been cultivated and produced intentionally for the first time in the 15th century, coffee spread to Europe quickly during the 17th century [53] and has become one of the most consumed beverages and thus traded commodities worldwide due to its flavor and psychostimulating properties [54,55]. As shown in Figure 3, except for two years of negative fluctuation (2019/20 and 2020/21), global consumption has been steadily growing over the last ten years (by 17.6% from 2012/13 to 2021/22) [56,57,58] and is forecasted to further grow [56,59].

Of the approximately 100 species of the genus *Coffea*, *C. arabica* and *C. canephora* Pierre ex A. Froehner are mainly used for coffee production and are therefore the most economically valuable [15,59]. Together, these two species contribute to ca. 99% of global coffee production [59,60]. After cultivation, the cherry-like coffee fruits, hence alternatively named coffee cherries, are usually harvested by hand-picking [39,54,61]. Once harvested, the coffee fruits are processed via a dry or wet method [61,62] and converted to green coffee beans. These are then roasted, ground, and after preparation with hot (or lately also cold) water, consumed as the commonly known coffee brew or beverage [39,54,61].

Besides green coffee beans, the economically most important coffee plant organ [63], several by-products accrue during cultivation and production [64,65]. For food production, the most promising of these by-products are coffee leaves, flowers, cherry husks or pulp (also known as cascara), parchment (the endocarp of the coffee fruit), silver skin (a thin tegument, also referred to as testa or epidermis, covering the coffee beans), and spent grounds (the residue or mass left after coffee brewing) [62]. Yet, coffee by-products are either, if at all, used as natural fertilizers or, in many cases, considered waste and discarded. However, since sustainable food production is of growing ecological importance, utilizing these by-products more efficiently appears highly advisable [65,66]. In fact, as concluded at the 2nd International Electronic Conference on Foods discussing coffee by-products as sustainable foods, they offer significant economic value [64]. Moreover, being rich in carbohydrates and/or fibers, proteins, and antioxidants, nutritional value has been attributed to them [67,68,69,70,71]. Therefore, dietary use as food may constitute an economically and nutritionally beneficial option [64]. Indeed, within the European Union (EU), several coffee by-products are currently undergoing approval procedures or have already been approved to be introduced to the EU market as so-called novel foods or traditional foods from a third country under Regulation (EU) 2015/2283 by the EU Commission. For detailed information on this, see [62,64,65]. The official definitions of novel food and traditional food from a third country can be found here [72]. Note, all authorized novel foods are listed in the annex of the Regulation (EU) 2017/2470, in the permanently updated Union List of Novel Foods [73].

Importantly, all of the above coffee by-products contain trigonelline, with levels varying from 0.40 to 62.6 g/kg (see Section 3). The authorization of coffee by-products to be marketed as food in the EU will therefore introduce additional dietary sources of trigonelline. This may lead to increased oral exposure of the general population in the EU. The aim of this review was therefore to assess the risk to human health from acute and chronic oral exposure to trigonelline from coffee and coffee by-products.

## 2. Literature Research

For this purpose, the electronic databases Google Scholar and PubMed were searched for available literature on trigonelline in coffee and coffee by-products. The strategy of the literature search consisted of three steps. To characterize trigonelline and its pharmacokinetics in humans, in the first step, it was specifically searched for original articles providing information on the biosynthesis, absorption, bioavailability, distribution, metabolism, elimination, and/or excretion of trigonelline in humans. This step also included gathering nutritional information on trigonelline and data to provide a brief background on coffee and coffee by-products. In the second step, the databases were searched for original articles on in vitro, in vivo, and/or human data on toxicological endpoints. For the risk assessment of acute and chronic exposure, a realistic worst-case scenario was assumed, estimating the maximum amount of trigonelline theoretically ingested from coffee and coffee by-products per day. For this, information on the trigonelline content and daily consumption of the respective matrices was collected in the third step of the literature search.

The search terms used included, amongst others, trigonelline, 1-methylpyridin-1-ium-3-carboxylate, coffee, coffee by-products, toxicity, adverse effects, exposure, and human. Only materials of *C. arabica* and *C. canephora* origin were included, as these are the economically and dietary most important coffee species in the EU [59]. Other species were excluded as they are of minor economic and dietary importance [60] and usually contain considerably lower trigonelline concentrations [74]. Studies not indicating the coffee species investigated were likewise excluded. All retrieved articles or hits deemed suitable for this work, based on the information provided in their abstracts, were entered into a bibliographic management software, and subsequently processed in detail individually (n = 190). As the literature search revealed a shortage of data, suitable articles were considered regardless of their publication date. Only studies accessible in the full text were used for this review.

## 3. Estimation of Human Oral Exposure

### 3.1. Trigonelline Contents of Coffee and Coffee By-Products

Data on the trigonelline contents are comparatively scarce. For coffee silver skin and spent grounds, only one study was found, with Wang et al. reporting to be the first to detect trigonelline in coffee silver skin [75]. Table 2 lists the trigonelline contents as reported by the accessible studies meeting the inclusion criteria outlined in Section 2. For easier comparison, the original values were additionally converted to g/kg.

As shown, the trigonelline contents vary greatly from 0.2 to 63 g/kg (factor ~200) depending on multiple factors, for instance: country of origin [89], coffee species and cultivar [74,90], part of the coffee plant [9], age of the plant part [19,76,77,79], applied industrial production processes [6,91,92], used preparation methods [93,94], water content of the sample [84], pH condition of the sample media [95], and analytical method used [95].

Due to the distinct water solubility and thermolability of trigonelline, concentrations in green coffee beans were shown to be markedly reduced during industrial (a) washing and blanching, leading to losses via leaching [6], and (b) roasting and steaming, causing losses via thermally induced decomposition [14,40,96]. Consequently, profoundly lower concentrations were found in roasted than in green (i.e., unroasted) coffee beans. Numerous studies have demonstrated that both water- and heat-induced reduction depend on the time-temperature profile applied, with 15–90% of trigonelline being degraded with increasing processing time and/or temperature [6,17,40,95,97,98,99]. Fast roasting is therefore associated with higher trigonelline contents than slower roasting approaches [74]. Furthermore, decaffeination was shown to lower the trigonelline content of coffee beverages by up to 17.1% [74,96,98]. Besides industrial processes, the characteristics of the method used for preparing the final beverage have proven to be a crucial factor. For instance, Ziefuß et al. revealed that hot-brew coffee variants contain noticeably less trigonelline than their cold-brewed counterparts [93]. When espresso coffees were prepared from *C. arabica* and *C. canephora* blends using a regular espresso machine set at 92 °C and 9 bar, the trigonelline content fell by 92.3% from the shortest (0–10 s) to the longest extraction time (36–40 s) [87]. Considering these trigonelline-reducing influences, it is not surprising that the lowest concentrations are observed in coffee beverages and the remaining spent grounds. According to Wu et al., coffee beverages from *C. canephora* usually contain about one-third less trigonelline than those from *C. arabica* [90], which concurs with the findings of Jeszka-Skowron and colleagues [96]. The highest trigonelline content was interestingly reported to be present in coffee flowers [44]. Contrarily, according to other authors, the trigonelline content of coffee flowers does not exceed 20 g/kg [80,81], thus being about three times lower than reported by Abreu Pinheiro et al. [44]. This discrepancy may be partially explained by the different analytical methods used and the age of the flowers analyzed. As described by Monteiro et al., the trigonelline content of coffee flowers correlates inversely with the flower’s age and decreases by up to 24.1% accordingly [76]. This correlation has also been observed in coffee leaves, with contents naturally declining as the leaf matures and senesces [18,79].

### 3.2. Theoretical Maximum Daily Intake of Trigonelline

In order to estimate the oral exposure of the EU general population to trigonelline from coffee beverages, knowledge of the daily consumption of coffee beverages and the mass of ground, roasted coffee beans weighed-in per volume of water is necessary. Yet, most sources use the term “cup of coffee” without specifying the amount of ground beans used in the preparation of the final drink. Thus, according to Romualdo et al., the most reliable data are represented by estimates of the annual consumption of ground beans per capita (which are typically expressed in kilograms) [55]. However, from such data, actual consumption habits regarding daily intake of coffee beverages can only be vaguely derived. Thus, data on the daily consumption of coffee beverages in the EU were adopted from the comprehensive European Food Consumption Database of the European Food Safety Authority (EFSA), as published in the 2018 International Agency for Research on Cancer (IARC) monograph, Drinking Coffee, Mate, and Very Hot Beverages [100], selecting the highest estimate reported. Accordingly, a daily maximum of 1914 mL of coffee was drunk per capita in Denmark (sex and year unspecified) [100].

No data on the daily consumption of coffee by-products per capita in the EU were available. Therefore, a realistic worst-case scenario was assumed. As reviewed by Klingel et al., coffee leaves, flowers, cherry husks or pulp, green coffee beans, and silver skin may be used to make tea [62]. Recently, an infusion of coffee leaves from *C. arabica* and/or *C. canephora* has been authorized for marketing by the EU Commission under Regulation (EU) 2015/2283 [72,101]. Thus, the consumption of coffee by-products as tea infusion was viewed as a realistic scenario. Within this scenario, the following assumptions were made: For the preparation of one serving, 2 g of the respective coffee by-product and 200 mL of cold water are used. Except for the cold water, this is in concordance with the procedure described by Steger et al., who used hot water [94]. In this scenario, cold water is used because hot water may induce thermal degradation of trigonelline, thus lowering the possible amount that could be ingested (see Section 3.1). Further, since trigonelline is highly soluble in water (see Section 1), a complete extraction into the tea infusion was assumed to occur.

Data on daily consumption of tea infusions were obtained from the German National Consumption Study II (Nationale Verzehrsstudie II). This is the largest epidemiological study conducted in Germany to collect representative information on the dietary habits and food consumption of the German general population. The study was coordinated by the Max Rubner-Institut (MRI) and conducted from 2005 to 2007. Using computer-assisted personal interviews, diet history interviews covering the last four weeks, written questionnaires, and 24 h dietary recalls, data from 19,329 participants (thereof, 8923 males and 10,406 females) aged 14–80 years were collected [102]. Different foods were considered and divided into certain main groups and subgroups, with tea infusions assigned to the main group “alcohol-free beverages” and the two subgroups (a) “coffee and green/black tea” and (b) “herbal/fruity tea”. Here, subgroup (a) was excluded since the proportions between coffee and green/black tea are not specified, making a reasonable estimate impossible. Thus, subgroup (b) was considered solely. Accordingly, a maximum (95th percentile) of 1300 g of herbal and/or fruity tea was consumed by females daily; males drank a maximum of 800 g per day [103]. In the sense of the worst-case scenario outlined, the amount consumed by females was considered. For the purpose of the present work, it is assumed that 1300 g is equivalent to 1300 mL.

Note, for coffee parchment (possible use in food preservatives or antioxidants [62]) and spent grounds (possible use in bakery products [62]), an exposure estimate is not feasible due to lacking data on the proportion to which parchments and spent grounds might be used in certain foods. Roasted coffee beans are assumed to be utilized exclusively for making coffee beverages, as is typically the case in the EU [93]. Therefore, a separate estimation was not performed. Table 3 shows the results of this exposure estimation.

By far, the highest theoretical maximum daily intake of trigonelline results from coffee beverages. The theoretical amount ingested via coffee by-products is significantly lower (factor 17–194) and is negligible in relation to coffee beverages.

## 4. Absorption, Distribution, Metabolism, and Excretion

The pharmacokinetics of trigonelline have repeatedly been studied in humans [17,22,23,24,25,26,92,104,105,106,107,108]. Accordingly, absorption and oral ingestion are strongly correlated [22,23,24,26,105,107,108]. Since plasma concentrations responded rapidly as early as 15 min after administration, absorption is proposed to start in the stomach [26,92]. Peak plasma concentrations were reached at 3.00–8.48 h after administration [92,108], suggesting trigonelline to be predominantly absorbed in the small intestine [92].

No data on oral bioavailability in humans are available. In pathogen-free Sprague–Dawley rats, however, the absolute bioavailability of trigonelline administered orally via a methanol extract of coffee beans amounted to 64.42% [109].

In humans, differences in plasma concentrations between the sexes have been observed. Whilst Lang et al. found no differences in plasma concentrations between male and female participants within the first 2–4 h after ingestion, sex-dependent differences were observed in later phases, as peak plasma concentrations of females were 16.3% higher and reached considerably later (females: 3.17 h, males: 2.29 h) [26]. Coinciding with these results, Bresciani and colleagues found significantly higher C_min_, C_avg_, C_max_, and AUC_0–24_ in females in relation to males [108]. These intersexual differences may be attributable to the lower body mass and blood volume of the recruited females, resulting in less dilution of trigonelline in the blood [26,108]. Further, plasma trigonelline was shown to be 20% lower in fasting than in non-fasting participants and increased by 9% per decade of age [107]. It is unclear whether smoking affects plasma levels. While it has been described that plasma concentrations of former and current smokers were 24 and 48% higher, respectively, than those of participants who had self-reportedly never smoked in their life [107], another study showed no statistically significant effect of smoking [108].

The volume of distribution (V_d_), calculated on the basis of data from 13 volunteers, was described as comparatively small at 123 L [26] (for context: Holford and Yim [110] define the V_d_ as very small at 10 L and large at 500 L), suggesting that trigonelline remains primarily in the blood [17,22,23,26]. Yet, in transgenic mice, trigonelline entered the cerebral cortex [111], implying the ability of this compound to cross the blood-brain barrier.

Little is known about its metabolism in humans. Upon oral administration, four potential methylation and oxidation products were detected in plasma and urine: *N*-methylpyridinium, *N*-methylnicotinamide, *N^1^*-methyl-4-pyridone-5-carboxamide, and *N^1^*-methyl-2-pyridone-5-carboxylic acid [105,108,112]. Moreover, trigonelline may be transformed to further pyridinium compounds via ring fission [105].

Up to 64.3% of the ingested amount is excreted unchanged via the urine within the first 8 h after ingestion [26,92,105], and additionally, about 10% as *N^1^*-methyl-2-pyridone-5-carboxylic acid [105], leaving the fate of the remaining ~30% unclear. Considering the good water solubility of trigonelline, it is likely that the vast majority of this proportion may also be excreted via the urine at later stages without undergoing any prior chemical changes [17,105]. As with absorption, excretion and oral ingestion are positively correlated [17,22,23,24,25,26,105,106,108]. It is noteworthy that trigonelline accumulated in human plasma at markedly elevated levels for an extended period of time following repeated ingestion of coffee beverages [26,92], due to the long biological elimination half-life of around 5.5 h [92,108]. Thus, plasma elimination and subsequent renal excretion occur relatively slowly [26,92]. Therefore, complete clearance of plasma trigonelline is unlikely in humans who habitually consume coffee several times a day [92]. Yet, plasma concentrations decreased back to baseline levels when coffee consumption was discontinued [108].

## 5. Nutritional Information

Trigonelline, as a constituent of green coffee beans, decomposes readily when heated at roasting temperatures above 180 °C [14], giving rise to various volatile derivatives such as pyrroles, alkylpyridines, and phenolic compounds, thereby influencing the flavor and aroma of coffee products. Additionally, nicotinic acid is produced as one of two non-volatile derivatives following *N*-demethylation (the second being *N*-methylpyridinium as a decarboxylation product) [17,39,92]. Serving as a precursor for essential nicotinic acid (niacin or vitamin B_3_) in coffee beans, trigonelline thus has nutritional value [8,17,39]. However, it is unclear whether intact trigonelline ingested via foods is metabolized to nicotinic acid in the human body and thus acts as a provitamin. Metabolic transformation to nicotinic acid is only assumed to potentially occur [26].

The German Nutrition Society (DGE) recommends a daily intake of 16 mg of niacin for male adults aged 19–24 years [113]. According to Caprioli et al., espresso coffee from *C. canephora* contains up to 10.3 mg of niacin per serving (25 mL) [87]. Thus, a single serving of espresso coffee could contribute to roughly 64% of the recommended daily intake of niacin. Yet, no significant increase in plasma niacin levels was observed in six female and seven male healthy volunteers following a bolus dose of 350 mL of coffee beverages from *C. arabica* ingested within 5 min [26]. To answer the question of whether trigonelline contributes considerably to the recommended daily intake of niacin further studies are needed.

## 6. Toxicological Information

Historically, trigonelline has been investigated focusing on beneficial effects on human health. Studies examining toxic or adverse effects are scarce and mostly date back many years to several decades. Toxicological studies have primarily been performed using cell cultures or laboratory animals. Thus, few human data exist.

### 6.1. Acute Toxicity

Trigonelline did not exert any cytotoxic effects in concentrations up to 100 µmol/L in human neuroblastoma SK-N-SH cells [114], human hepatocellular carcinoma (Hep3B) cells [115], human immortalized dermal keratinocytes (HaCat) and human foreskin fibroblasts (Hs68) [45] after treatment for six days, 48 and 24 h, respectively. In fact, in a recent study, trigonelline protected UV-B radiation-exposed Hs68 cells against photodamage by restoring the physiological function of the endoplasmic reticulum, decreasing oxidative stress, and positively modulating calcium homeostasis and apoptosis in a dose-responsive manner (10, 25, 50, and 100 µmol/L) [45]. It also did not alter the morphology of Madin–Darby bovine kidney cells and African green monkey kidney cells (Vero cells) at concentrations of 1.6 µg/mL each after 48 h of exposure [42].

In an animal study from 1946, the first of its kind to examine the acute effects of trigonelline according to the literature search, Brazda and Coulson administered trigonelline subcutaneously to rats (strain and sex not provided). As a result, a dose of 5000 mg/kg body weight (bw) was found to be lethal (LD_50_) [116]. This observation has been corroborated twice: first, by Mishkinsky et al., who determined the same LD_50_ after oral administration to female albino Sabra rats [117]; and second, by Aswar et al., who did not notice alteration of behavior or increased mortality in mice (strain and sex not provided) after administration of doses up to 5000 mg/kg bw via unspecified route [118]. No papers reporting other LD_50_ values were found. Additionally, no human data on acute toxicity were uncovered.

As no data from human studies are available, the acute toxicity of isolated trigonelline was predicted in silico, among other toxicological endpoints (see Section 6.2, Section 6.3 and Section 6.4), using the freely available computational platform ProTox-II [119]. With a moderate prediction accuracy of 69.26%, the results for the acute oral toxicity of trigonelline were negative. The methods used in the in silico predictions are described in detail in the Model Info section of the ProTox-II platform. Additionally, a short description of this tool is provided here [120].

### 6.2. Subchronic Toxicity

Three animal studies were identified with designs appropriate for the investigation of subchronic toxicity. In cats (strain and sex not provided), trigonelline administered at doses up to 3500 mg/kg bw for 62–70 days did not induce any visible effects [121]. Additionally, when albino Sabra rats were fed 50 mg/kg bw of trigonelline daily for 21 days and sacrificed after 42 days, no changes in the weight of the thyroid, thymus, kidneys, liver, adrenals, ovaries, and uterus were observed [117]. In a more recent study from 2014, Folwarczna et al. exposed ovariectomized (i.e., estrogen-deficient) and non-ovariectomized, three-month-old female Wistar rats orally to 50 mg/kg bw of trigonelline daily for 28 days. No adverse effects were noticed in the non-ovariectomized rats. In contrast, a significantly increased weight of the uterus and L4 vertebra was observed in the ovariectomized, estrogen-deficient rats. Estrogen deficiency led to the onset of osteoporotic changes of the skeleton. Oral administration of trigonelline aggravated these skeletal changes by worsening the mechanical properties and mineralization of cancellous bones. The mechanism by which these adverse effects were produced is unknown [122].

No reports on subchronic effects in humans have been published. However, the results of the in silico predictions performed using ProTox-II did not suggest any signs of subchronic toxicity of isolated trigonelline [119].

### 6.3. Genotoxicity and Mutagenicity

The first in vitro experiments on the potential mutagenicity of trigonelline were conducted by Fung et al. in 1988. The authors showed that pure trigonelline did not produce any mutagenic activity in an L5178Y TK^+/−^ mouse lymphoma assay and several *Salmonella typhimurium* test systems (i.e., Ames tests) [10]. Contrarily, in the *S. typhimurium* assays conducted by Wu and colleagues, heated trigonelline, applied to the plates (a) alone and (b) combined with certain amino acids caused substantial amounts of revertants, especially when in binary combination with serine (12,740 revertants/mmol) or threonine (11,270/mmol). Of all the 13 single compounds examined in that study in strain TA98, heated trigonelline applied to the plates alone produced the most revertants (8160/mmol) and exhibited the highest mutagenicity [7]. Table 4 summarizes the results of these studies.

The approaches of the *S. typhimurium* assays in both studies differed strongly (different strains, metabolic activation, and doses (see Table 4)); thus, comparing the results is hardly possible. A major difference is that, in contrast to Fung and colleagues, Wu et al. mimicked the roasting process of green coffee beans by heating the samples at 250 °C for 20 min during sample preparation, analyzing trigonelline in an approach that more realistically simulates common exposure scenarios, since green coffee beans are not known to be consumed raw in the EU [62]. Before analysis, samples were given time to settle to room temperature. As discussed by Wu et al., the heating of trigonelline may have led to the formation of free radicals and compounds involved or entirely responsible for the mutagenic effects observed. Thus, the production of revertants could not be ascribed to trigonelline unambiguously [7]. Since only these two studies were found and their results are equivocal, further efforts are needed to conclude whether trigonelline is genotoxic and/or mutagenic.

However, in the in silico toxicity model computations performed in the present work, trigonelline was predicted not to be mutagenic (*p* = 0.94) [119].

### 6.4. Carcinogenicity

Trigonelline has been examined for its potential role in cancer in several in vitro experiments. In only one study was it shown to have effects that could increase the risk of tumor formation, as it significantly induced the proliferation of MCF-7 breast cancer cells in a dose-responsive manner. Noticeably increased total cell numbers were observed at considerably low concentrations of 10 pmol/L, with the most potent proliferative effect yielded at 100 pmol/L. The growth-promoting effect was due to trigonelline binding the estrogen receptors (ER) of the MCF-7 cells in a way similar to estradiol, thereby altering ER-mediated transcription. As discussed by the authors, trigonelline has no structural similarity to estradiol or phytoestrogens. Thus, the authors suggested that trigonelline did not bind the active ligand binding site of the ER, but probably another domain, changing the receptors’ conformation and enabling binding of unspecified transcriptional co-regulators. Hence, trigonelline exerted sound estrogenic activity [123]. This is a significant finding as increased estrogenic activity—either induced by pathological overexpression of ER or mediated by estrogen-like compounds encountered, for instance, via the diet—has been shown to be an important factor in some types of cancer (e.g., breast, colorectal, endometrial, and ovarian cancer [124]), as it stimulates cell growth and proliferation, promoting cancer initiation [125]. In contrast, trigonelline did not yield estrogenic activity in female Wister rats (see Section 6.5) [118], suggesting no trigonelline-induced estrogen-related carcinogenicity in vivo. Moreover, in the in silico carcinogenicity models computed in this work, no carcinogenic effects of trigonelline were predicted (*p* = 0.66) [119].

Conversely, several in vitro studies have interestingly demonstrated anticarcinogenic effects instead [115,126,127,128,129]. According to Hirakawa and colleagues, trigonelline inhibited the reactive oxygen species (ROS)-potentiated invasion of pretreated AH109A rat ascites hepatoma cells at comparatively much higher concentrations of 2.5–40 µmol/L by means of scavenging ROS [127]. Anticarcinogenic effects were also exerted by significantly inhibiting the migration of Hep3B cells at 100 µmol/L. The cell migration was hampered by a complex cascade of effects. Briefly, trigonelline decreased the phosphorylation of the nuclear factor E2-related factor-2 (Nrf2), a key transcription factor in the gene expression of several antioxidative and xenobiotic-detoxifying enzymes, at serine-40, downregulating its activity and thus the expression of catalase, glutathione peroxidase, and superoxide dismutase [115,128,129]. Consequently, cancer cell survival was possibly decreased by the impaired response to oxidative stress [130], hampering the migration of cancerous cells. Trigonelline also inhibited the gene expression of zinc-dependent matrix metalloproteinase-7 (MMP-7) [115], an endopeptidase that degrades extracellular matrix, reduces cancer cell adhesion, and inhibits apoptosis, thereby suppressing tumor migration and progression [131,132]. Furthermore, trigonelline-mediated inhibition of MMP-7 led to downregulation of the Raf/ERK/Nrf2 signaling pathway, which is believed to play a crucial role in many mechanisms associated with the pathogenesis of various types of cancer [115]. Furthermore, 24 h exposure to 100 µmol/L of trigonelline slightly downregulated the expression of K-RAS, an important proto-oncogene in the progression of colorectal carcinoma, in caco-2 human colon carcinoma cells treated with coffee extracts from *C. arabica* [133]. Sharma et al. showed that trigonelline (50 µmol/L) reduced the expression of peroxisome proliferator-activated receptor-γ and sterol regulatory element-binding protein-1 in palmitic acid-induced fatty liver disease model human hepatoma (HepG2) cells, suggesting that trigonelline protects against the development of hepatocellular carcinoma by decreasing fatty-acid-mediated lipotoxicity [134].

In total, the above data suggest rather anticarcinogenic effects for trigonelline, mainly by means of positively modulating gene expression. To fully elucidate its role in cancer, appropriate longitudinal in vivo studies are necessary. Yet, such studies are highly resource-consuming and have not been conducted to date.

Additionally, while no epidemiological studies are available on trigonelline as an isolated compound, there are ample studies available on coffee consumption, in which trigonelline is one of the major constituents. In a major review of the evidence on coffee, the IARC suggested that there is inadequate evidence in humans for the carcinogenicity of drinking coffee. There is evidence suggesting a lack of carcinogenicity of drinking coffee in humans for cancers of the pancreas, liver, female breast, uterine endometrium, and prostate. Inverse associations with drinking coffee have been observed with cancers of the liver and uterine endometrium [100]. Hence, it may be questionable whether the current evidence would justify the time and resources needed to further study the long-term effects of trigonelline regarding carcinogenicity.

### 6.5. Reproductive Toxicity and Teratogenic Effects

Except for one in vivo study from 2009, no data on reproductive toxicity or teratogenic effects are available. In that study, Aswar et al. aimed to investigate the effects of trigonelline on the estrous cycle and fertility of female Wistar rats. To examine effects on the estrous cycle (endpoint: vaginal cornification), the authors administered doses of 75 mg/kg bw of trigonelline orally to ovariectomized, immature female Wister rats weighing 55–60 g twice on the first two days and once on the third and fourth day of the treatment. Afterwards, vaginal smears were collected and microscopically searched for cornified or nucleated epithelial cells. As no such cells were noticed, it was concluded that trigonelline did not exert estrogenic activity. The same dose was fed daily for seven days to non-ovariectomized, pregnant female Wistar rats (150–200 g bw) to investigate abortifacient activity. No significant differences in the number of implants and litters were observed between treated and control animals. Litters survived and experienced normal growth. Thus, trigonelline did not exhibit abortifacient activity [118].

### 6.6. Neurotoxicity

Over 50 years ago, Gill et al. investigated trigonelline in a tincture of *Cannabis sativum* L. and concluded that its neurological agonistic activity is approximately 10^5^ times lower than that of the physiological neurotransmitter acetylcholine. No neurotoxic effects were observed in that in vitro study [135]. Half a century later, current knowledge is still consistent with these findings, as trigonelline has not been shown to exhibit adverse effects on the nervous system in any of the test systems used. Instead, several studies have demonstrated neuroprotective properties [111,114,136,137,138,139,140,141,142,143]. These have been reviewed in detail elsewhere in 2021 [52].

In brief, doses up to 10 µmol/L ameliorated the atrophy of neuronal dendrites and axons in amyloid β-peptide-treated female, hemizygous transgenic 5XFAD Alzheimer’s disease mice model [111]. Additionally, trigonelline (30 µmol/L) improved functional neurite outgrowth in human neuroblastoma SK-N-SH cells after treatment for six days [114,141]. Furthermore, daily oral administration of 500 mg/kg bw for 15 days was demonstrated to improve memory retention in six-week-old amyloid β-peptide-treated male ddY mice [136]. Thus, trigonelline is believed to improve cognition in Alzheimer’s disease model laboratory animals [141,142]. Furthermore, it has been discussed that trigonelline may protect against cerebral ischemia [140] by positively modulating neuronal spike frequency [143], stimulating the release of dopamine [139], competitively inhibiting γ-aminobutyric acid (GABA) A-receptors [138], and weakly hampering acetylcholine esterase [137].

Yet, due to the lack of clinical reports and epidemiological studies, it is impossible to conclude on potential protective effects against neurological diseases in humans.

### 6.7. Immunotoxicity and Allergenicity

No allergenic properties of trigonelline have been reported. In fact, when six-week-old female BALB/c mice were topically exposed to concentrations up to 200 µmol/L on repeated days during a 16-week in vivo treatment, no signs of allergic reactions were noted in the daily physical examinations of the animals [45]. Additionally, an aqueous solution of powdered dried coffee husks from *C. arabica*, which has been shown to contain rather low levels of trigonelline (see Table 2), did not induce any allergic reactions in skin-prick tests performed on 40 human individuals [144].

In contrast, trigonelline has interestingly been shown to protect against allergic reactions in RBL-2H3 cells and six-week-old female BALB/c model mice. In both experiments, trigonelline suppressed antigen-stimulated mast cell degranulation in a dose-dependent manner through several mechanisms. These mechanisms included inhibition of secretion and phosphorylation of several mediators involved in the cellular degranulation signaling pathways: Concentrations up to 10 mmol/L inhibited the secretion of β-hexosaminidase by up to 59.4% and significantly reduced the activating phosphorylation of PLCγ1, PI3K, and Akt. In addition, the formation of microtubules, which are crucial for the intracellular movement of granules and their fusion with the cell membrane to enable granule release or degranulation, was hampered [145].

### 6.8. Other Adverse Effects

No other adverse effects of trigonelline have been documented.

## 7. Regulatory Information

No limit values, such as tolerable or acceptable daily intake levels, have been derived for trigonelline. Therefore, to our knowledge, no specific regulations are currently in force in the EU or other parts of the world.

## 8. Risk Assessment

### 8.1. Limitations

As studies on the bioactivity of trigonelline have focused on beneficial health effects, toxicological data are scarce. There is particularly a shortage of human data, with longitudinal epidemiological and clinical studies altogether lacking. Most of the accessible in vitro studies used supraphysiological doses exceeding the amount of trigonelline that could realistically be ingested through the consumption of coffee and coffee by-products. In vivo studies, on the other hand, are often characterized by mechanistic limitations [55] and cannot be extrapolated to human scale without uncertainty due to interspecies differences. Furthermore, the exposure estimates made in the present work are specifically rather vague for coffee by-products, as little is known about the trigonelline content and level of consumption of these matrices.

### 8.2. Acute Oral Exposure of Trigonelline

The only toxicological threshold experimentally established to date for acute oral exposure to trigonelline that can be used in a risk assessment is the oral LD_50_ of 5000 mg/kg bw derived from rat studies. Using the method of Gold et al. [146], the LD_50_ values were extrapolated to the lower one-sided confidence limit of the benchmark dose (BMDL)10 values, assuming linear behavior (as no other information on dose-response is available), to obtain a more conservative toxicological threshold (see also Lachenmeier and Rehm [147] for remarks on extrapolation when only LD_50_ values are available for risk assessment).

For an average person weighing 70 kg, the extrapolated BMDL_10_ corresponds to an intake of 34.3 g of trigonelline per day. Thus, an average person would have to consume 4.8 L of coffee beverages (Table 5), the matrix shown to contribute most to the acute exposure (see Table 3), in a single day to reach this value. This level of consumption is deemed unrealistic as it exceeds the maximum daily coffee consumption reported for member states of the EU (see Table 3) by factor 2.5 and is beyond any volume of coffee beverages that can rationally be expected to be consumed by a person throughout one day. Therefore, no toxic or adverse effects upon acute oral exposure to trigonelline through consumption of coffee beverages are to be expected.

Since tea infusions prepared from coffee by-products contain significantly lower levels of trigonelline than coffee beverages (Table 5), as estimated in the assumed worst-case scenario, harmful effects are less likely to occur after acute oral exposure from these matrices, as the corresponding volume of beverage that would need to be consumed over the course of a day to meet the BMDL_10_ is completely unrealistic.

### 8.3. Chronic Oral Exposure of Trigonelline

Individuals habitually consuming coffee beverages and/or coffee by-products daily are chronically exposed to trigonelline.

As presented in Section 4, repeated ingestion of coffee beverages led to an accumulation of trigonelline in human plasma and to markedly elevated plasma levels, which, on the premise of continued regular coffee consumption, are unlikely to be completely cleared due to the long biological half-life of trigonelline. Considering its low octanol-water partition coefficient and volume of distribution, on the other hand, trigonelline is not expected to accumulate in human body tissues. In fact, no trigonelline-induced toxic effects have been observed in animals or humans in studies on reproductive toxicity, teratogenic effects, neurotoxicity, immunogenicity, and allergenicity. Taking into account the limitations mentioned in Section 8.1 and the ambiguous results of the genotoxicity and mutagenicity studies, the current data do not allow for a reasonable, soundly substantiated risk assessment for chronic oral exposure to trigonelline. Additional studies on potential toxic effects are needed because the current toxicological database is insufficient. Additionally, further efforts are recommended to address open questions, such as the trigonelline-mediated aggravation of osteoporotic skeletal changes shown in a subchronic toxicity study in estrogen-deficient rats. No other study on this subject was found, and it is unclear whether humans diagnosed with estrogen deficiency or related conditions may experience similar unfavorable effects from chronic oral exposure to trigonelline.

For a comprehensive risk assessment, the presence of trigonelline in other dietary or food sources and their role in human oral exposure must also be considered.

On the other hand, the EFSA has evaluated the safety of several trigonelline-containing coffee by-products as an obligatory step in the authorization process of novel foods and traditional foods from a third country in the EU. In all cases, no safety concerns were raised [144,148,149]. Furthermore, coffee consumption is characterized by a long history, having been consumed in Europe since the 17th century [53]. Despite this long history, no reports of toxic effects attributed to trigonelline are known. In this respect, trigonelline appears to be rather safe in concentrations ingested via coffee and coffee by-products, which would also question the priority of this compound for further toxicological studies.

## 9. Conclusions

In this review, oral exposure to trigonelline from coffee and coffee by-products was assessed for health-related risks based on current pharmacokinetic and toxicological knowledge and available consumption data for the EU general population. No signs of adverse effects following acute exposure were identified. However, due to the scarcity of toxicological data and the lack of epidemiological and clinical studies, it is not possible to draw a well-founded conclusion on the safety of chronic exposure to trigonelline orally ingested from the concerning matrices. The current state of knowledge, however, suggests that chronic exposure to trigonelline in contents typical for coffee and coffee by-products may also be unproblematic in terms of human health.

## Figures and Tables

**Figure 1 molecules-28-03460-f001:**
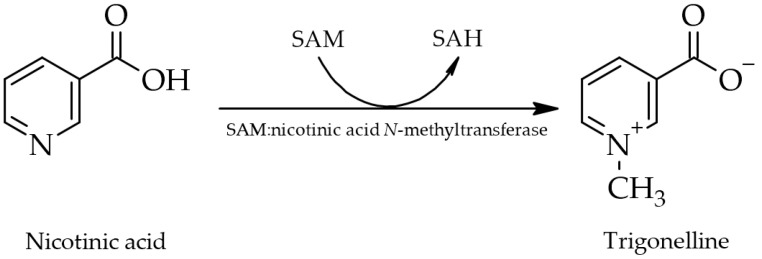
Biosynthesis of trigonelline in plants upon methylation of nicotinic acid via SAM:nicotinic acid *N*-methyltransferase (SAM, *S*-adenosyl-*L*-methionine. SAH, *S*-adenosyl-*L*-homocysteine) [27,28,29].

**Figure 2 molecules-28-03460-f002:**
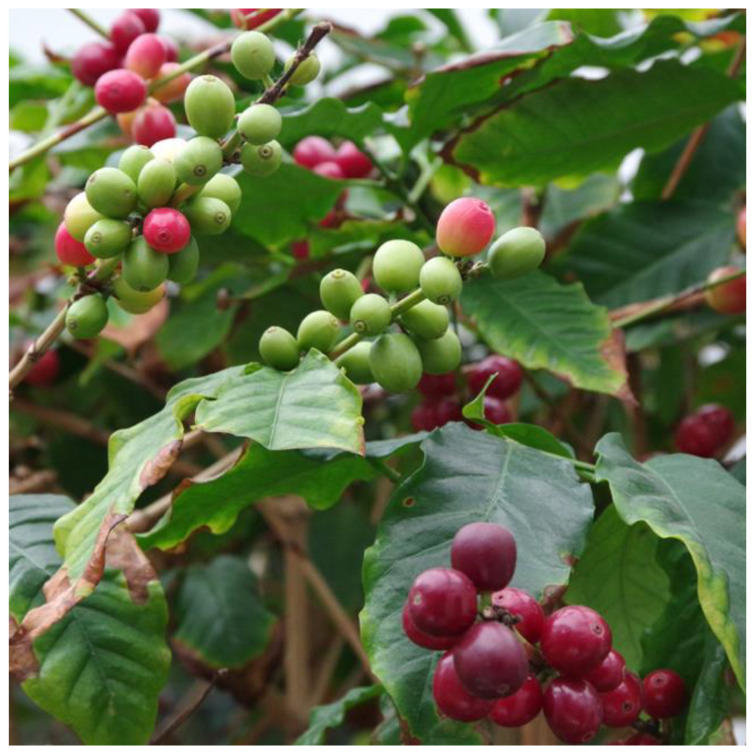
Coffee fruits in different stages of ripeness.

**Figure 3 molecules-28-03460-f003:**
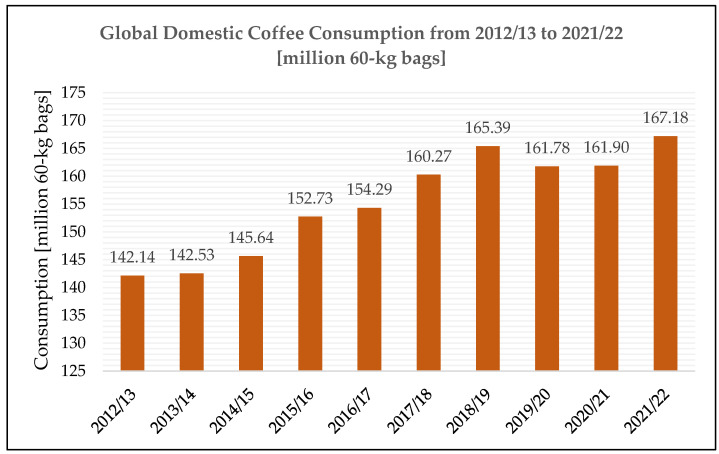
Global domestic coffee consumption from 2012/13 to 2021/22 in million 60 kg bags [56,57,58].

**Table 1 molecules-28-03460-t001:** Physicochemical properties of trigonelline [2].

Physicochemical Property	Information
Molecular weight	137.14 g/mol
Octanol–water partition coefficient (log K_OW_)	−2.53

**Table 2 molecules-28-03460-t002:** Trigonelline contents of coffee and coffee by-products of *C. arabica* and *C. canephora* origin.

Matrix	Trigonelline Content	Coffee Species	Reference	Converted Trigonelline Content (g/kg)
Leaves	2.955–11.718 mg/g DW	A	[76]	3.0–11.7
	3.93–6.84 mg/g	A	[77]	3.9–6.8
	4.47 ± 0.13 g/kg DW	A	[78]	4.5
	8.2–28.3 µmol/g FW	A	[18]	1.1–3.9
	4.157–7.879 mg/g DW	R	[76]	4.2–7.9
	4.2–6.0 mg/g DW	R	[79]	4.2–6.0
			Range	1.1–11.7 (min–max)
Flowers	378.1–5278.6 mg/100 g DW	A	[44]	3.8–52.8
	429.1–6258.3 mg/100 g DW	R	[44]	4.3–62.6
	1092.8 mg/100 g DW	R	[80]	10.9
	755–1965 mg/100 g	A, R	[81]	7.6–19.7
			Range	3.8–62.6 (min–max)
Cherry Husks/Pulp	285.58–542.8 mg/100 g DW ^1^	A	[6]	2.9–5.4 ^1^
	550.45–558.02 mg/kg DW	A	[82]	0.6–0.6
			Range	0.6–5.4 (min–max)
Parchment ^2^	1.24–1.36% DW	A	[83]	12.4–13.6
	120.16–246.21 mg/100 g DW ^2^	A	[6]	1.2–2.5 ^2^
			Range	1.2–13.6 (min–max)
Green coffee beans	1.52–2.9 g/100 g DW	A	[4]	15.2–29.0
	8.8–27.6 mg/g DW	A	[79]	8.8–27.6
	0.88–1.77% DW	A	[16]	8.8–17.7
	9.1–15.9 g/kg	A	[20]	9.1–15.9
	16.8–71.2 µmol/g	A	[18]	2.3–9.8
	55.67 µmol/g FW	A	[17]	7.6
	52.5 ± 0.8 µmol/g FW	A	[13]	7.2
	0.98–1.32% w/w DW	A	[84]	9.8–13.2
	547–991 mg/100 g	A	[14]	5.5–9.9
	7.5–34.2 mg/g DW	R	[79]	7.5–34.2
	3.08 g/100 g DW	R	[4]	30.8
	0.75–1.24% DW	R	[16]	7.5–12.4
	7.5–12.0 g/kg	R	[20]	7.5–12.0
			Range	2.3–34.2 (min–max)
Roasted coffee beans	5.25–7.48 g/kg DW	A	[85]	5.3–7.5
	3.69–4.81 mg/g	A	[14]	3.7–4.8
	41.82 ± 2.1 µmol/g FW	A	[17]	5.7
	3.08–5.54 g/kg DW	R	[85]	3.1–5.5
			Range	3.1–7.5 (min–max)
Silver skin	3.65% DW	A	[75]	36.5
				36.5
Coffee beverages	1.9–7.2 mg/mL	A	[86]	1.9–7.2
	50.74–72.70 mg/25 mL (espresso)	A	[87]	2.0–2.9
	0.19–0.33 mg/mL	A	[88]	0.2–0.3
	2310 nmol/mL	A	[26]	0.3
	21.99–49.44 mg/25 mL (espresso)	R	[87]	0.9–2.0
			Range	0.2–7.2 (min–max)
Spent coffee grounds	0.4–2.4 mg/mL	A	[86]	0.4–2.4
			Range	0.4–2.4 (min–max)

^1^ Samples comprised 80% husks and pulp and 20% parchment. ^2^ Samples comprised 28% husks and pulp and 72% parchment. A, *C. arabica*. R, *C. canephora*. FW, fresh weight. DW, dry weight. If neither FW nor DW is indicated, the referenced study does not provide this information.

**Table 3 molecules-28-03460-t003:** Theoretical maximum daily intake of trigonelline via coffee and coffee by-products.

Matrix	Trigonelline Content per Serving (g/200 mL) ^1^	Consumption of Tea Infusions or Coffee Beverages (mL/day)	Theoretical Maximum Daily Intake (g/day)
Leaves	0.023	1300	0.152
Flowers	0.125	1300	0.814
Cherry Husks/Pulp	0.011	1300	0.070
Green coffee beans	0.068	1300	0.445
Silver skin	0.073	1300	0.475
Coffee beverages	1.44	1914	13.8

^1^ For coffee by-products, a preparation of tea infusions with 2 g of coffee by-product per 200 mL of cold water was assumed. For further explanation, see text. For coffee beverages, the assumption of weighing-in 2 g of, in this case, ground, roasted coffee beans, was not made as data for the final beverage are available (see Table 2). To account for the outlined worst-case scenario, for each matrix the highest trigonelline content as reported in Table 2 was used in the calculations.

**Table 4 molecules-28-03460-t004:** Results for trigonelline in different genotoxicity and mutagenicity test systems.

Test System	Strains	Metabolic Activation	Dose of Trigonelline	Result	Ref.
L5178Y TK^+/−^ mouse lymphoma assay	-	With and without 0.5 mL of S9 from Aroclor-1254-induced male Fischer 344 rats	Up to 7429 µg/mL	Negative	[10]
*S. typhimurium* assay	TA1535 TA1537 TA1538 TA98 TA100	With and without 0.5 mL of S9 from Aroclor-1254-induced male Fischer 344 rats and Syrian golden hamsters	Up to 10,000 µg/plate	Negative	
*S. typhimurium* assay	TA98	With 0.5 mL of S9 from chlorophene-induced rat liver	1000 µmol	Positive	[7]
			Plus 1000 µmol of a single amino acid:		
			Alanine	Positive	
			Arginine	Positive	
			Cysteine	Positive	
			Cystine	Negative	
			Lysine	Positive	
			Phenylalanine	Positive	
			Proline	Positive	
			Serine	Positive	
			Threonine	Positive	
			Tryptophan	Negative	
			Valine	Positive	
			Plus 200,000 µmol/L of glucose	Positive	
*S. typhimurium* assay	TA98	With 0.5 mL of S9 from chlorophene-induced rat liver	Mix I ^1^	Positive	[7]
			Mix II ^2^	Positive	
		Without 0.5 mL of S9 from chlorophene-induced rat liver	Mix I	Positive	
			Mix II	Positive	
	YG1024	With 0.5 mL of S9 from chlorophene-induced rat liver	Mix I	Positive	
			Mix II	Positive	
		Without 0.5 mL of S9 from chlorophene-induced rat liver	Mix I	Toxic	
			Mix II	Toxic	
	YG1029	With 0.5 mL of S9 from chlorophene-induced rat liver	Mix I	Negative	
			Mix II	Negative	
		Without 0.5 mL of S9 from chlorophene-induced rat liver	Mix I	Positive	
			Mix II	Positive	

^1^ Mix I consisted of 1000 µmol trigonelline, 100 µmol glucose, and a mixture of alanine, arginine, aspartic acid, asparagine, glutamic acid, glycine, histidine, isoleucine, leucine, lysine, phenylalanine, proline, serine, threonine, tyrosine, valine, and tryptophan. ^2^ Mix II consisted of 1000 µmol trigonelline, 100 µmol glucose, and a mixture of aspartic acid, asparagine, and glutamic acid. The molar percentages of the amino acids are indicated in the original paper [7]. For simplicity, details on the respective test procedures are not shown, but can be read in the referenced studies.

**Table 5 molecules-28-03460-t005:** Estimated beverage volume (L) for reaching oral BMDL_10_ of 490 mg/kg bw of trigonelline.

Matrix	Beverage Volume (L) per Day for Reaching Oral BMDL_10_ = 490 mg/kg bw ^1^
Leaves	293
Flowers	55
Cherry husks/pulp	635
Green coffee beans	100
Silver skin	94
Coffee beverages	4.8

^1^ For explanation on assumed preparation of servings considered in the calculations, see Section 3.2. An estimate of BMDL_10_ is obtained from LD_50_ (5000 mg/kg bw) by division by 10.2 using Method B of Gold et al. [146].

## Data Availability

No new data were created or analyzed in this study. Data sharing is not applicable to this article.

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
