# Peer review of "Risk Assessment of Trigonelline in Coffee and Coffee By-Products"

_molecules, 2023, doi:10.3390/molecules28083460_

Round 1
Reviewer 1 Report
This paper has made a study on Risk Assessment of Trigonelline. Although the author has done some works, but there are many aspects need to be improved. I suggest the paper can be accepted after the following points can be revised.
1. The literature data used in the article should be described in Section 2, and the rationality of the coverage of literature data should be explained.
2. In Section 8, Risk Assessment of Trigonelline as the main task has not been discussed specifically enough.
Author Response
This paper has made a study on Risk Assessment of Trigonelline. Although the author has done some works, but there are many aspects need to be improved. I suggest the paper can be accepted after the following points can be revised.
- The literature data used in the article should be described in Section 2, and the rationality of the coverage of literature data should be explained.
RESPONSE: The strategy used in the literature search was added in section 2. Also, the description of the literature data in section 2 was improved and a rationale on the coverage of literature was included.
- In Section 8, Risk Assessment of Trigonelline as the main task has not been discussed specifically enough.
REPSONSE: Thank you for your comment! However, in the opinion of the authors (and as described in section 8, and especially in subsection 8.2.), it is not possible to discuss this matter more specifically or in more detail due to the shortage and/or lack of toxicological data. From the authors’ point of view, the risk assessment is based on all the existing data that could be considered in a reasonable risk assessment. To emphasize the reference of the risk assessment to trigonelline, the words “…of Trigonelline” has been added to the headings of subsections 8.2. and 8.3. as well as to the heading of Table 5. The authors also edited the paragraph on lines 1127-1131 (of the revised manuscript). Referring to the tea infusions made from coffee by-products, the conclusion that “the corresponding volume of beverage that would need to be consumed over the course of a day to meet the BMLD10 is completely unrealistic” was added to this paragraph.
We would like to thank Reviewer 1 for reviewing our paper and the helpful comments!
Reviewer 2 Report
This paper provides a review of the available toxicological information on trigonelline for assessing its potential health risks if it is consumed in other food products besides coffee. Although the toxicology data are too sparse for a strong risk assessment and the authors show that the amount of trigonelline in other food products is negligible compared to coffee, and they state that the safe use of coffee for a long period of time calls into question the need for further toxicological testing, I think it is still important to get this information into the published literature.
The paper overall could use some editing for English and there are also a few things that appear to be mistakes or stray words, and some places where things could be clarified or the text could be shorter/more streamlined. I have put specific comments for these things below.
Line 30: The word should be "first" and not "firstly"
Line 57: As this paper will probably be read by human health toxicologists who do not know plant physiology, please define "nyctinasty" and "nodulation."
Line 76: Use the word "except" rather than "excepting"
Lines 78-79: The end of this sentence needs editing and would read better as: "...years (by 17.6% from 2012/13 to 2021/22 [55-57] and is forecasted to grow further [55, 58]."
Line 83: There is a stray "L" after arabica
Line 84: There is stray text "Pierre ex A. Froehner" after canephora
Lines 86-94: The text here is too long and not all of the information is necessary. The sentence in lines 86-87 about the species of minor economic importance can be deleted. Line 89 can be shortened to end the sentence with "...are usually harvested by hand-picking." The rest of the paragraph can be shortened to just "Once harvested, the coffee fruits are processed and converted to green coffee beans."
Line 106: The sentence should end with "...constitute an economically and nutritionally beneficial option."
Line 137: The word "respectively" is not needed
Line 143: There is a stray "L" after arabica and "Pierre ex A. Froehner" after canephora
Line 150: It should be "analytical method used"
Line 162: It should be "has proven to be" and not "proofed"
Line 174: "the one" should be deleted
Line 175: It should be "by" and not "with"
Lines 237-239: It should be "By far, the highest theoretical maximum...lower, and is negligible in relation to coffee beverages."
Line 280: Delete "sort of"
Line 320: Delete "instead"
Line 321: Should this be "Madin-Darby" and not "Madine Darby"?
328-330: Specify the dose of the Aswar study.
Line 357: Use "conducted" instead of "self-reportedly performed"
Line 380: It should be "may have led"
Line 388: It should be "potential role in cancer" because in vitro studies cannot examine actual carcinogenesis (tumor formation)
Lines 389-390: Cell proliferation is not a carcinogenic effect (it is not tumor formation). It happens often without forming a tumor. Please rephrase this sentence.
Line 403: There should be a comma after proliferation
Line 407: It should be "no prediction of carcinogenic effects" because in silico modeling is only predictive, not indicative of.
Line 461: Delete "in that experiment either"
Line 464: There is a stray "L" after sativum
Lines 492-498: This sentence is too long, it should be broken up into more than one sentence for ease of reading.
Line 512: Delete "manifold"
In the references, there is a stray "L" after arabica and/or a stray "Pierre ex A. Froehner" after canephora at the following lines: 856, 861, 970, 978, 979, 982.

Author Response
This paper provides a review of the available toxicological information on trigonelline for assessing its potential health risks if it is consumed in other food products besides coffee. Although the toxicology data are too sparse for a strong risk assessment and the authors show that the amount of trigonelline in other food products is negligible compared to coffee, and they state that the safe use of coffee for a long period of time calls into question the need for further toxicological testing, I think it is still important to get this information into the published literature.
The paper overall could use some editing for English and there are also a few things that appear to be mistakes or stray words, and some places where things could be clarified or the text could be shorter/more streamlined. I have put specific comments for these things below.
Line 30: The word should be "first" and not "firstly"
RESPONSE: The word was corrected accordingly. It now reads “first”.
Line 57: As this paper will probably be read by human health toxicologists who do not know plant physiology, please define "nyctinasty" and "nodulation."
RESPONSE: The definitions of both nyctinasty and nodulation were added in parentheses directly after the respective term. In this context, the references 34 and 35 were added to the paper.
Line 76: Use the word "except" rather than "excepting"
RESPONSE: The wording was changed accordingly.
Lines 78-79: The end of this sentence needs editing and would read better as: "...years (by 17.6% from 2012/13 to 2021/22 [55-57] and is forecasted to grow further [55, 58]."
RESPONSE: The sentence was edited according to your suggestion.
Line 83: There is a stray "L" after arabica
RESPONSE: The stray “L” was deleted.
Line 84: There is stray text "Pierre ex A. Froehner" after canephora
RESPONSE: Here, C. canephora Pierre ex A. Froehner is mentioned for the first time in the text. Therefore, “Pierre ex A. Froehner” was added to give the full scientific name, which includes the name of the first discoverer. Therefore, the authors prefer not to delete it.
Lines 86-94: The text here is too long and not all of the information is necessary. The sentence in lines 86-87 about the species of minor economic importance can be deleted. Line 89 can be shortened to end the sentence with "...are usually harvested by hand-picking." The rest of the paragraph can be shortened to just "Once harvested, the coffee fruits are processed and converted to green coffee beans."
RESPONSE: The sentence in lines 86-87 was deleted. As suggested, the sentence in line 89 was shortened to end with “…are usually harvested by hand-picking.” The rest of the paragraph was shortened to “Once harvested, the coffee fruits are processed via a dry or wet method [62, 63] and converted to green coffee beans.” The authors consider the information on the processing method (dry or wet) to be useful and have therefore decided not to delete it.
Line 106: The sentence should end with "...constitute an economically and nutritionally beneficial option."
RESPONSE: Thank you for comment! The sentence was edited and corrected accordingly.
Line 137: The word "respectively" is not needed
RESPONSE: The word “respectively” was deleted.
Line 143: There is a stray "L" after arabica and "Pierre ex A. Froehner" after canephora
RESPONSE: The “L” and “Pierre ex A. Froehner” were deleted.
Line 150: It should be "analytical method used"
RESPONSE: The word order was changed and corrected accordingly.
Line 162: It should be "has proven to be" and not "proofed"
RESPONSE: This grammatical mistake was corrected accordingly.
Line 174: "the one" should be deleted
RESPONSE: As suggested, the expression “the one” was deleted.
Line 175: It should be "by" and not "with"
RESPONSE: This grammatical mistake was corrected accordingly.
Lines 237-239: It should be "By far, the highest theoretical maximum...lower, and is negligible in relation to coffee beverages."
RESPONSE: The sentence was edited and rephrased according to the suggestion. It now reads, “By far, the highest theoretical maximum daily intake of trigonelline results from coffee beverages. The theoretical amount ingested via coffee by-products is significantly lower (factor 17–194), and is negligible in relation to coffee beverages.”
Line 280: Delete "sort of"
RESPONSE: As suggested, the expression “sort of” was deleted.
Line 320: Delete "instead"
RESPONSE: Thank you for your comment” The word “instead” was deleted.
Line 321: Should this be "Madin-Darby" and not "Madine Darby"?
RESPONSE: The typo was corrected. It now reads “Madin-Darby”.
328-330: Specify the dose of the Aswar study.
RESPONSE: The dose administered by Aswar et al. (up to 5000 mg/kg bw) was added to the sentence.
Line 357: Use "conducted" instead of "self-reportedly performed"
RESPONSE: As suggested, “self-reportedly performed” was deleted and replaced by “conducted”.
Line 380: It should be "may have led"
RESPONSE: The word “have” was added.
Line 388: It should be "potential role in cancer" because in vitro studies cannot examine actual carcinogenesis (tumor formation)
RESPONSE: The word “potential” was added.
Lines 389-390: Cell proliferation is not a carcinogenic effect (it is not tumor formation). It happens often without forming a tumor. Please rephrase this sentence.
RESPONSE: The sentence was rephrased to “In only one study was it shown to have effects that could increase the risk of tumor formation, as it significantly induced proliferation of MCF-7 breast cancer cells in a dose-responsive manner”.
Line 403: There should be a comma after proliferation
RESPONSE: The missing comma after “proliferation” was added.
Line 407: It should be "no prediction of carcinogenic effects" because in silico modeling is only predictive, not indicative of.
RESPONSE: The sentence was rephrased. It now reads, “Moreover, in the in silico carcinogenicity models computed in this work, no carcinogenic effects of trigonelline were predicted (P = 0.66) […].”
Line 461: Delete "in that experiment either"
RESPONSE: As suggested, the above expression was deleted.
Line 464: There is a stray "L" after sativum
RESPONSE: Here, Cannabis sativum L. is mentioned for the first time in the text. The abbreviation “L.”, which stands for Linnaeus, the Swedish botanist who formalized the binomial nomenclature or system of naming organisms, was included to give the full scientific name of this plant. Therefore, the authors prefer not to delete it.
Lines 492-498: This sentence is too long, it should be broken up into more than one sentence for ease of reading.
RESPONSE: The concerning sentence was rephrased and broken up into three sentences to improve readability,
Line 512: Delete "manifold"
RESPONSE: As suggested, the word “manifold” was deleted.
In the references, there is a stray "L" after arabica and/or a stray "Pierre ex A. Froehner" after canephora at the following lines: 856, 861, 970, 978, 979, 982.
RESPONSE: However, in all of the above cases, the “L.” and/or the “Pierre ex A. Froehner” are part of the original title of the respective reference. Therefore, in the opinion of the authors, they should not be deleted, since the titles should be cited in their original form.
Thank you for your comments on my review. My review contained many examples or grammar or typo issues because that was mostly what I found to be the issue with the manuscript. However, I will be sure to not focus on these types of edits in future reviews, knowing that all manuscripts will undergo copyediting and proofreading. I apologize if the few more substantial points of my review are hard to identify among all of the grammar suggestions.
I noted in my review that while this manuscript is a bit limited by the lack of much toxicity data to undertake a risk assessment, it is still good to get this information in the published literature. I found no major issues with the methodology (given that there was little information to work with) or the conclusions, as the authors acknowledged the limitations of the evaluation in an appropriate manner. My only additional comment upon looking at the paper again is that the Introduction should be shortened. It currently has a lot of information and some of this is not directly relevant to the evaluation in the paper and can confuse the reader about the point of the paper. For example, the information about trigonelline's discovery and synthesis in lines 40-50 and Figure 1 is extraneous and can be deleted. Similarly, the function of trigonelline in plants described in lines 55-59 is also not necessary as this paper is focused on human risk assessment and not plants. I had already provided comments on removing some text to streamline the paragraph in lines 82-94, but I think that this paragraph could use even further shortening to just note which are the two most economically important species of coffee plant (since they are the focus of the risk assessment).
RESPONSE: Thank you for your comments!
Regarding lines 40-59: Reviewer 3 highlights the background information provided as positive, as it gives the reader a clear overview of trigonelline. The authors agree on this and have therefore decided not to delete the information provided in these lines.
Regarding lines 82-94: The paragraph in lines 82-94, however, has been streamlined as suggested in the relevant comments above. To further shorten this paragraph, the following part of the sentence was also deleted: “…, with 87.4 million 60 kg-bags of C. arabica and 78.8 million 60 kg-bags of C. canephora coffee produced in 2021/22 [55]”.
We would like to thank Reviewer 2 for reviewing our paper and the valuable comments!
Reviewer 3 Report
This review comprehensively assessed the risk of trigonelline, a pyridine alkaloid presents in coffee and coffee by-products, to human health. This manuscript is well-organized and easy to understand. The background and detail information about trigonelline such as the estimation of human oral exposure, nutritional information, and various toxicological information and comparison are included in this manuscript to provide the readers a clear overview on trigonelline. Although current toxicological knowledge and human data of trigonelline are still limited, in silico prediction were performed in this review to provide further estimation. Based on the available data, the conclusions still give useful and practical risk assessment results that chronic exposure to trigonelline in contents typical for coffee and coffee by-products may be unproblematic for human health.
Some minor issues are listed as follows:
1. Line 36-37: “Comprising a carboxylate (-CO2-) and quaternary ammonium (R4N+) moiety, it is considered a betaine.” This description is confusing because trigonelline should not contain quaternary ammonium (R4N+) moiety?
2. Line 98: “……parchment, silver skin, and spent grounds” It will be clearer if additional information about the corresponding plant parts of these by-products can be added into this sentence.
3. Line 266: “……comparatively low at 123 L” It will help the readers to understand this description if the authors can provide more information on the low distribution volume (What is the low range or high range of distribution volume).
4. Line 354: “……did not suggest any no signs of subchronic……” “no” should be removed from this sentence?
5. Could the authors provide brief information of lymphoma assay and S. typhimurium assay in section 6.3?
Author Response
This review comprehensively assessed the risk of trigonelline, a pyridine alkaloid presents in coffee and coffee by-products, to human health. This manuscript is well-organized and easy to understand. The background and detail information about trigonelline such as the estimation of human oral exposure, nutritional information, and various toxicological information and comparison are included in this manuscript to provide the readers a clear overview on trigonelline. Although current toxicological knowledge and human data of trigonelline are still limited, in silico prediction were performed in this review to provide further estimation. Based on the available data, the conclusions still give useful and practical risk assessment results that chronic exposure to trigonelline in contents typical for coffee and coffee by-products may be unproblematic for human health.
Some minor issues are listed as follows:
- Line 36-37: “Comprising a carboxylate (-CO2-) and quaternary ammonium (R4N+) moiety, it is considered a betaine.” This description is confusing because trigonelline should not contain quaternary ammonium (R4N+) moiety?
RESPONSE: In the literature, trigonelline is described as a compound containing a quaternary ammonium moiety. However, since the reviewer found this information to be confusing, the authors decided to delete this information or sentence altogether (especially since this information is not critical to the text).
- Line 98: “……parchment, silver skin, and spent grounds” It will be clearer if additional information about the corresponding plant parts of these by-products can be added into this sentence.
RESPONSE: Additional information about these three by-products were added to the sentence. It now reads, “For food production, the most promising of these by-products are coffee leaves, flowers, cherry husks or pulp (also known as cascara), parchment (the endocarp of the coffee fruit), silver skin (a thin tegument, also referred to as testa or epidermis, covering the coffee beans), and spent grounds (the residue or mass left after coffee-brewing) [63].”
- Line 266: “……comparatively low at 123 L” It will help the readers to understand this description if the authors can provide more information on the low distribution volume (What is the low range or high range of distribution volume).
RESPONSE: The low and high ranges of volume of distribution, as defined by Holford and Yim, have been added to the sentence in parentheses as additional information. Reference 110 has been added to the paper.
- Line 354: “……did not suggest any no signs of subchronic……” “no” should be removed from this sentence?
RESPONSE: The word “no” has been removed from the sentence.
- Could the authors provide brief information of lymphoma assay and S. typhimuriumassay in section 6.3?
RESPONSE: Both assays are standard tests in the field of human toxicology. Therefore, the authors did not consider it necessary to provide detailed information on these assays. As described in lines 473-474 (of the revised manuscript), details of the respective test procedures can be found in the referenced studies. To clarify what is meant by S. typhimurium assay, the alternative (possibly better known) name of this assay, the Ames test, has been added to the text in parentheses. Also, lines 478-480 (of the revised manuscript) have been edited to emphasize the differences in the approaches of the S. typhimurium assays between the two studies.
We would like to thank Reviewer 3 for reviewing our paper and the constructive comments!
Round 2
Reviewer 1 Report
The author has revised all comments, and I suggest that the article be accpeted.